

# Comparable behaviour of ring and little fingers due to an artificial reduction in thumb contribution to hold objects

Banuvathy Rajakumar and Varadhan SKM

Department of Applied Mechanics, Indian Institute of Technology Madras, Chennai, India

## ABSTRACT

**Background**. The human hand plays a crucial role in accomplishing activities of daily living. The contribution of each finger in the human hand is remarkably unique in establishing object stabilization. According to the mechanical advantage hypothesis, the little finger tends to exert a greater normal force than the ring finger during a supination moment production task to stabilize the object. Similarly, during pronation, the index finger produces more normal force when compared with the middle finger. Hence, the central nervous system employs the peripheral fingers for torque generation to establish the equilibrium as they have a mechanical advantage of longer moment arms for normal force. In our study, we tested whether the mechanical advantage hypothesis is supported in a task in which the contribution of thumb was artificially reduced. We also computed the safety margin of the individual fingers and thumb.

**Methodology**. Fifteen participants used five-finger prismatic precision grip to hold a custom-built handle with a vertical railing on the thumb side. A slider platform was placed on the railing such that the thumb sensor could move either up or down. There were two experimental conditions. In the "Fixed" condition, the slider was mechanically fixed, and hence the thumb sensor could not move. In the "Free" condition, the slider platform on which the thumb sensor was placed could freely move. In both conditions, the instruction was to grasp and hold the handle (and the platform) in static equilibrium. We recorded tangential and normal forces of all the fingers.

**Results**. The distribution of fingertip forces and moments changed depending on whether the thumb platform was movable (or not). In the free condition, the drop in the tangential force of thumb was counteracted by an increase in the normal force of the ring and little finger. Critically, the normal forces of the ring and little finger were statistically equivalent. The safety margin of the index and middle finger did not show a significant drop in the free condition when compared to fixed condition.

**Conclusion**. We conclude that our results does not support the mechanical advantage hypothesis at least for the specific mechanical task considered in our study. In the free condition, the normal force of little finger was comparable to the normal force of the ring finger. Also, the safety margin of the thumb and ring finger increased to prevent slipping of the thumb platform and to maintain the handle in static equilibrium during the free condition. However, the rise in the safety margin of the ring finger was not compensated by a drop in the safety margin of the index and middle finger.

Corresponding author
Varadhan SKM, skm@iitm.ac.in

## INTRODUCTION

Many of our daily activities, such as holding a pen or lifting a cup, demand the use of our hand. An object held in the hand has to be maintained stationary, i.e., in static equilibrium for preventing tilt and slip. Studies have used a prehension handle to examine how forces of fingers and thumb are controlled during grasping. It is known that there will be a change in the distribution of fingertip normal forces whenever there is a change in the tangential force due to vertical lifting of the object followed by expected or unexpected changes in the load of the object (*Cole & Abbs, 1988*). Also, it has been shown that there is a tight temporal coupling between the normal and tangential forces during the point to point and cyclic arm movements with various types of grips (*Flanagan & Tresilian, 1994*). In addition to this, frictional state of the grasping surface was altered with different materials (like sandpaper, suede and silk) to exhibit a change in the tangential force which would be accompanied by the scaling of normal forces (*Johansson & Westling, 1987*). Similarly, the fingertip forces get redistributed among the individual fingers and thumb due to the change in the direction and magnitude of the external torque imposed to the handle (*Shim, Latash & Zatsiorsky, 2003*). In a study by Aoki and colleague's (*2006*) friction at the object-digit interface of the thumb and index finger side was altered to examine the force distribution. Tangential forces on the smoother side were found to be lower compared to the rougher side, whereas the normal forces were modulated based on the frictional condition.

Studies on prehension stability have also shown that during tasks involving the production of pronation or supination moment, fingers with larger moment arms (i.e., index and little fingers) tend to produce a greater share of normal force compared to the fingers with the shorter moment arms (i.e., middle and ring fingers). In our study, as per the handle design, the finger force sensors were placed at two cm away from each other. Therefore, the index and little fingers have longer moment arms for normal force when the fingers are placed on the sensors. This led to the formulation of the mechanical advantage hypothesis (referred to as MAH henceforth) (cf. *Buchanan, Rovai & Rymer, 1989*; *Prilutsky, 2000*). Zatsiorsky and colleagues conducted an experiment using five-finger prehension handle with a horizontal beam attached at the bottom of the handle (*Zatsiorsky, Gregory & Latash, 2002*). In each trial, a specific mass was suspended on the beam at a specific distance from the center of the handle. The task was to maintain the static equilibrium of the handle by producing either pronation or supination moment. They found that the moment arm of the finger had a monotonic non-linear relationship with the force produced by that finger. Index finger exerted greater normal force compared to the middle finger during the pronation moment production. Likewise, little finger produced greater normal force compared to the ring finger during the supination moment production. A similar handle was employed in the study by *Shim, Latash & Zatsiorsky (2005a)*, although the external torque acted in the direction perpendicular to the plane of grasp in that study. The task was to hold the handle in static equilibrium by performing radial or ulnar deviation to counteract the external torque. Normal force exerted by the peripheral fingers were greater than the normal force magnitude produced by central fingers to compensate the external torque. In both of these studies, external torque was introduced by means of

suspending load on the beam and individual fingers produced compensating moment to counterbalance the external torque. Also, MAH was tested in the grasping task that involves cyclic motion of the handle in the presence of external load at different positions (*Gao, Latash & Zatsiorsky, 2006*). The hypothesis was supported in all the above studies.

One common objective of researchers is to address how the central nervous system chooses a specific force pattern when changes are induced to the five finger prehension handle held in static equilibrium. Five finger prehension stability was examined when a change was introduced to the entire width of the handle (*Zatsiorsky, Gao & Latash, 2006*) or individual digit width other than the thumb in horizontal (*Slota, Latash & Zatsiorsky, 2012*) and individual digit placement (other than the thumb in the vertical direction) (*Solnik, Zatsiorsky & Latash, 2014*). In real life objects like a handheld portable radio, retractable ballpoint pens and certain models of pipette controller, a vertical tuner (or slider) is provided at the thumb side of the object to control the functionality. Proper orientation and positioning of the object are necessary for a stable grasp. The question of how object stabilization is achieved in such objects to compensate for the change in the moment caused due to the vertical unsteadiness of the thumb slider has not been addressed in the literature. In our study, a slider was positioned on the thumb side of the handle. In this way, we artificially reduced the contribution of thumb in holding the handle. Although our handle design does not closely resemble any of these objects, the core idea of providing an unsteady platform for the thumb emerged by observing the working of these objects. As a result of the reduction in the thumb contribution to hold objects, tangential force of thumb reduces. This, in turn, caused a decrease in the clockwise moment produced by the thumb. In response to this change, a counteracting supination moment has to be produced to establish handle stabilization. As a preliminary step, we aimed to address how object stabilization is achieved during static holding (with the thumb static). As per the MAH, we expected that the little finger would produce greater normal force compared to the ring finger to overcome the effect caused due to unsteady thumb platform. Thus, we hypothesized that the little finger with the larger moment arm would exert greater normal force in comparison to the ring finger with the shorter moment arm when the thumb contribution was artificially reduced.

Secondly, with regard to the drop in the load contribution of the thumb, there would be an increment in the tangential force of the virtual finger (VF). Virtual finger is an imaginary finger whose output is equal to the collective summation of the mechanical output from index, middle, ring and little fingers (*Iberall, 1987*). Since index and middle fingers are considered as independent and strong (*MacDermid et al., 2004*), we expected that they would share greater tangential force, which should be accompanied by the increase in the normal force. However, such an increase in the normal forces of radial fingers would cause a further tilt in the counter-clockwise direction. Therefore, to avoid such tilt, the normal forces of the radial fingers should remain unchanged, resulting in the drop of safety margin of the same. Hence, we hypothesized that the safety margin of the index and middle finger would drop when the thumb platform was free to slide in comparison to the thumb platform kept fixed. Furthermore, for a clear interpretation of the sequential local changes in the individual fingertip forces, we have employed the notion of chain

effects (*Shim, Latash & Zatsiorsky, 2003*; *Shim, Latash & Zatsiorsky, 2005a*; *Zatsiorsky, Gao & Latash, 2003*; *Shim, Latash & Zatsiorsky, 2005b*; *Niu, Latash & Zatsiorsky, 2009*; *SKM et al., 2012*). Chain effects refer to a sequence of local cause–effect adjustments which were necessitated either mechanically or as a choice made by the controller.

## MATERIALS AND METHODS

### Participants

Fifteen young healthy right-handed male volunteers (mean $\pm$ standard deviation Age: $25.6 \pm 2.7$ years, Height: $172.6 \pm 3.9$ cm, Weight: $73.3 \pm 9.6$ kg, Hand-length: $18.6 \pm 0.9$ cm, Hand-width: $8.7 \pm 0.3$ cm) participated in this experiment. Participants with any history of musculoskeletal or neurological illness were excluded.

### Ethical approval

The experimental procedures were approved by the Institutional ethics committee of IIT Madras (Approval number: IEC/2016/02/VSK-2/12. Full name of the committee that granted the approval: Institutional ethics committee of Indian Institute of Technology Madras). The experimental sessions were conducted in accordance with the procedures approved by the Institutional ethics committee of IIT Madras. Written informed consent was obtained from all participants before the start of the experiment.

### Experimental setup

We designed and built a vertically oriented prehension handle made of aluminium specifically for this study. The thumb side of this handle had a vertical railing. On this railing, we placed a slider platform such that it can move only in the vertical direction. The slider had ball bearings, and hence the friction between the slider and the railing was minimal ($\mu \sim 0.001$ to $0.002$). We stored the handle in a dust-free environment during non-use. Further, we regularly cleaned and lubricated the ball bearing between experimental sessions to ensure minimal friction. We used five 6-component force/torque sensors (Nano 17, Force resolution: 0.0125N, ATI Industrial Automation, NC, USA) to measure the fingertip forces and moments in the X, Y and Z directions. The thumb sensor was mounted on the slider platform. Hence the thumb sensor could freely move in the vertical direction, whereas the other finger sensors were fixed to the handle.

A laser displacement sensor (resolution, 5 $\mu$m; OADM 12U6460, Baumer, India) was mounted on a flat acrylic platform near the top of the handle on the thumb side. This sensor was used to measure the vertical displacement of the moving platform with respect to the geometric center of the handle. At the center of the handle frame, a thin horizontal solid line was drawn with a permanent marker to indicate the position at which the participants were required to maintain the slider in the free condition.

On top of the handle, an acrylic block extending in the anterior-posterior direction was placed. A spirit level was positioned on the participant side of the acrylic block. An electromagnetic tracking sensor (Resolution 1.27 $\mu$m, Static position accuracy 0.76 mm, Static angular orientation accuracy 0.15°, Model: Liberty Standard sensor, Polhemus Inc., USA) was placed on the other side of the acrylic block as shown in Fig. 1. Thirty analog

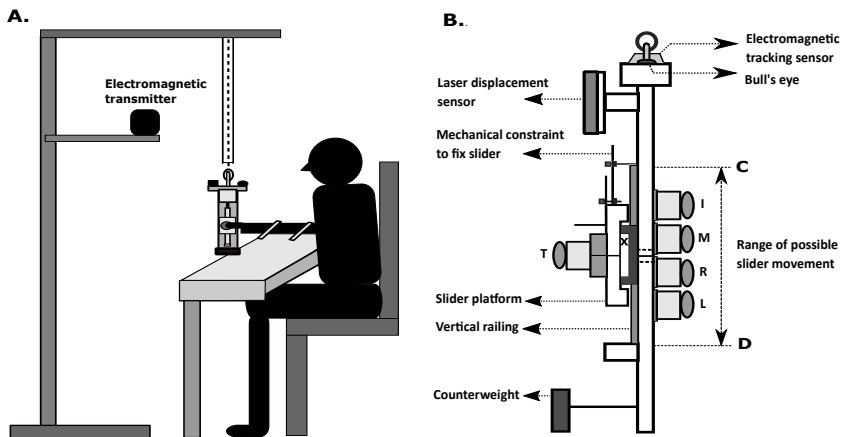

**Figure 1** **Experimental apparatus.** (A) Schematic diagram of a participant holding the handle Thumb side of the handle is shown. The entire handle setup was suspended from a wooden frame using nylon rope passing through a hollow PVC pipe. The PVC pipe allowed slight movement of the rope (and handle) but not undesirable large amplitude movement of the handle. The participant was required to lift the handle from its suspended position by two cm vertically, thus causing a slack of the nylon rope during the trial recording. The transmitter of the electromagnetic tracking system was placed a few cms away from the handle to avoid distortion. (B) Schematic diagram of the experimental setup ATI Nano 17 force sensors mounted on the handle frame (20 cm × 1 cm × 3 cm) to measure the forces of fingers (I-Index, M-Middle, R-Ring, Little-L, Th-Thumb). The geometric centre of the handle is represented by the symbol 'X on the slider. The centres of the force sensors (excluding the thumb) were placed at a distance two cm apart from each other. Two solid horizontal lines were drawn (one on the slider and the other on the handle frame between middle and ring fingers). In free condition, the slider platform can translate over vertical railing such that it can theoretically move from point C to point D. The maximum possible vertical displacement of the slider platform and hence the thumb sensor was seven cm. The horizontal distance between the grasping surfaces of the thumb and finger sensors was 6.5 cm. We covered the surface of all the force sensors with 100 grit sandpaper. Mass of the slider platform was 0.101 kg. The mass of the entire handle, including the slider was 0.535 kg. To bring the whole object center of mass close to the geometric center of the handle, a rectangular aluminium counter-weight of 0.035 kg was placed close to the bottom, on the thumb side of the handle.

signals from the force/torque sensors (5 sensors × 6 components) and single-channel analog laser displacement data were digitized using NI USB 6225 and 6002 at 16-bit resolution (National Instruments, Austin, TX, USA). This data was synchronized with six channels of processed, digital data from the electromagnetic tracker. The data were collected at 100 Hz.

## Experimental procedure

Participants washed their hands with mild soap and water before the beginning of the experiment. Friction experiment was performed first, followed by the Prehension experiment.

## Friction experiment

We designed a device that consists of a six-component force/torque sensor (Nano 25; ATI Industrial Automation, Garner, N.C) mounted on the top of the aluminium platform. The platform moved linearly with the help of a timing belt-pulley system powered by a servomotor (*Savescu, Latash, & Zatsiorsky, 2008*; *Park et al., 2014*). A customized LabVIEW

program was written for the data collection and to control the operation of the motor. Forearm and wrist movements of the participants were arrested by Velcro straps while a wooden block was placed underneath the participant's palm for the steady hand and finger configuration. Participants were instructed to produce a constant downward normal force of 6N for 3s to initiate movement of the servomotor. Visual feedback of the normal force was shown on the computer monitor for the participant. The platform moved at a speed of six mm/s away from the participant. Data was collected from the index and thumb finger only. One trial per finger was conducted. The friction coefficient was computed by dividing the tangential force and normal force at the time of slip.

## Prehension experiment

Participants were seated comfortably on a wooden chair with their forearm resting on the table. The right upper arm was abducted approximately 45° in the frontal plane, flexed 45° in the sagittal plane with the elbow flexed approximately about 90°. The natural grasping position can be achieved by supinating the forearm at 90°. The movements of the forearm and wrist were restricted by strapping them to the tabletop with Velcro.

The experiment involved a task that had two different conditions: "fixed" and "free". In the fixed condition, the vertical thumb slider was fixed securely using a mechanical constraint. This fixed position was such that the horizontal line passing through the center of the thumb sensor was precisely aligned with the solid horizontal line drawn at the center of the handle (i.e., between the center of the middle and ring finger sensors). In the free condition, this mechanical constraint was released so that the slider was free to vertically translate over the entire length of the vertical railing. Theoretically, the thumb sensor could move a maximum range of seven cm, approximately between the index finger and little finger. However, in the current study, we required the thumb platform to be maintained between middle and ring fingers. This was in addition to the requirement to maintain the handle in static equilibrium. The spirit level provided tilt feedback to the participant.

In both conditions, the task was to lift the handle vertically upward from the suspended position with their right hand to support the load of the handle with the fingers and thumb. The handle was required to be held in such a way that the fingertips' center approximately coincided with the center of each sensor. Eight participants performed free condition first followed by the fixed condition. The other seven participants performed fixed condition first followed by the free condition. The experimenter (but not the participant) could view the normal force of all fingers, slider's vertical displacement data, position and orientation of the handle. The trial started only after the participant held the handle in a stable manner and informed the experimenter to start. The participants were instructed to grasp and hold the handle vertical by maintaining the bubble in the bull's eye at the center throughout the trial. They were also instructed to lift the handle with all fingers in both conditions and position the horizontal line on the thumb platform matching the horizontal line drawn at the midline between the middle and ring finger in the free condition. Although the friction between the thumb platform and handle was low, it was not zero. The experimental task required some practice. Five practice trials were provided at the start of each condition (not analyzed). After practice, participants were able to follow the instruction and perform

the task successfully. Each experimental condition was conducted in a separate session. A rest period of one hour was provided between conditions. In each condition, thirty trials were recorded. Each trial lasted for ten seconds, with a minimum mandatory break period of thirty seconds between the trials. Additional rest was provided when the participants requested.

## Data analysis

The data was collected using a customized LabVIEW (LabVIEW Version 12.0, National Instruments) program, and offline analysis was performed in MATLAB (Version R2016b, MathWorks, USA). Force/Torque data were low-pass filtered at 15 Hz using second-order, zero phase lag Butterworth filter. We only considered the data between 2.5 and 7.5s (500 samples) for all the analyses to eliminate the start and end of trial effects.

## Normal force sharing (%)

Normal force sharing of the individual fingers (excluding the thumb) was expressed in terms of percentage by taking the average across 500 samples of each trial and then averaged across all trials and participants.

## Safety margin

Safety margin (SM) is the amount of extra normal force applied in addition to the minimally required normal force to avoid slipping of the handle. We computed SM for all fingers using the following equation (*Burstedt, Flanagan & Johansson, 1999*; *Pataky, Latash & Zatsiorsky, 2004*; *SKM et al., 2012*).

$$SM(t) = \frac{\left[ F_n - \frac{|F_t|}{\mu} \right]}{F_n} \tag{1}$$

where $\mu$ is the coefficient of friction between the finger pad and sandpaper, $t$ refers to the time course of 5 s, $F_n$ is the normal force, and $F_t$ is the tangential force applied to the object. SM was calculated with the corresponding friction coefficient value $\mu$ of each participant that was computed from the friction experiment data. The average friction coefficient of index and thumb computed across 15 participants were $0.9689 \pm 0.0054$ and $0.9745 \pm 0.0109$, respectively. For statistical analysis, Fisher's Z-transformed SM ($SM_z$) values were found by using the following equation.

$$SM_z = 0.5 * ln\left( \frac{1 + SM}{1 - SM} \right) \tag{2}$$

## Statistics

Statistical analyses were performed using R. Two-way repeated measures ANOVA were performed with the condition (2 Levels: Fixed and Free) X finger (5 Levels: Index, Middle, Ring, Little and Thumb) as factors for normal force, tangential force, $z$-transformed normal force sharing and safety margin. Sphericity test was performed on the data for all cases, and the number of degrees of freedom was adjusted using the Huynh-Feldt (H-F) criterion wherever required. Post-hoc pairwise comparisons were performed using Tukey test to

explore the significance within the factors. We also performed an equivalence test using Two One-Sided $T$-test (TOST) approach (*Lakens, 2017*), to check for equivalence of the tilt angles and safety margin between fixed and free conditions. Furthermore, the equivalence test (TOST) was performed (both for absolute and % values) between the normal forces of the ring and little fingers during the free condition to check if they are comparable. The TOST test approach on the six comparisons (tilt angles, safety margin of index, middle, little fingers, normal forces of ulnar fingers, % of normal force share of the ulnar fingers) were performed with the desired statistical power of 95% having a sample size of 15.

## RESULTS

### Task performance

The ideal performance of the task in both fixed and free condition would be to hold the object in static equilibrium. In the free condition, participants were also required to align the horizontal line on the slider to the horizontal line on the handle frame. Figure S1 (left and the right column) shows the time profiles of average normal force and average tangential force in the fixed and free condition. Note that the standard error of the means of the normal and tangential force of the individual fingers (excluding the thumb) in the free condition was found to be greater when compared to the fixed condition.

The tilt angles showed no statistically significant difference (fixed condition: Mean = 4.13°, SD = 2.31; free condition: Mean = 3.83°, SD = 1.82, t(28) = 0.395, $p = 0.696$, $d = 0.14$). The TOST procedure for the independent tilt angle samples was performed with the smallest effect size of interest (**SESOI = 1.31**) set as equivalence bounds (lower limit $\Delta_L = -1.31$ and upper limit $\Delta_U = 1.31$) obtained for the desired level of statistical power of 95%. The procedure revealed that the comparison was statistically equivalent (t(28) = $-3.192$, $p = 0.00174$) as the observed effect size (d) falls within the equivalence bounds. The amount of deviation of the center of thumb sensor from the marked position on the handle frame was calculated. It was computed by finding the absolute difference between the maximum and minimum values of the laser displacement data within each trial. This difference was then calculated for all trials, averaged, and then averaged across participants. During the free condition, the average deviation of the marked horizontal line on the slider from the horizontal line on the handle was $0.88 \pm 0.06$ mm.

### Changes in normal force and tangential force in the thumb and the other fingers

The normal force produced by the index finger (Mean = 1.65N, SEM = 0.15) in free condition task was found to be greater than the normal force produced by the index finger (Mean = 1.63N, SEM = 0.11) in fixed condition task. Hence, when plotted, the data of index finger normal force in fixed condition was found hidden under the plot of Index finger normal force during the free condition (see Fig. S1A). The average normal force of the middle, ring, little fingers and the thumb in free condition was significantly higher than that in the fixed condition. The average thumb tangential force in the free condition decreased significantly compared to the fixed condition. This decrease in the thumb's tangential force was compensated by the increase in the tangential force and normal force

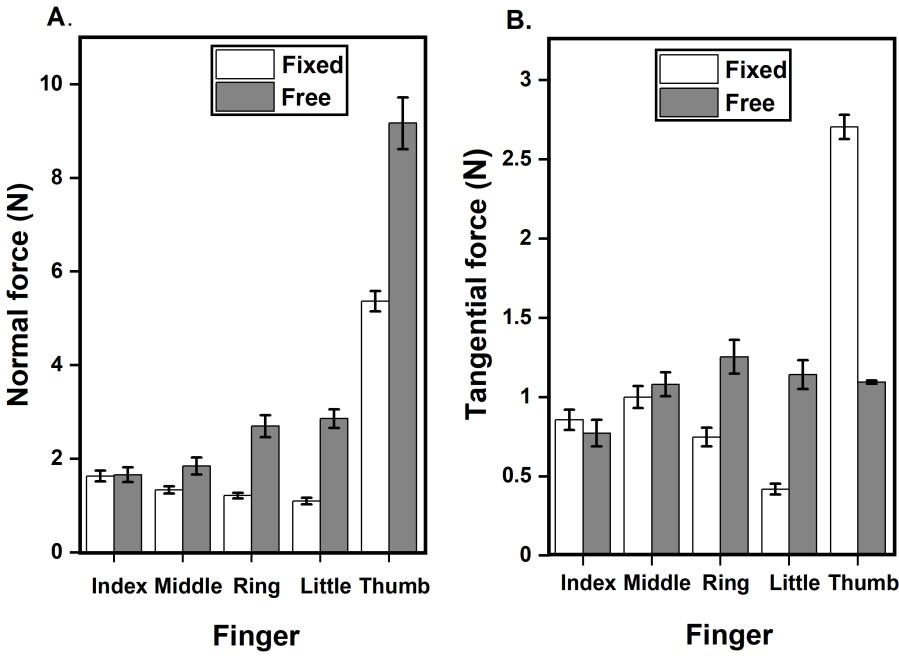

**Figure 2 Average normal force and tangential force of all fingers in different conditions.** (A) Average normal force of index, middle, ring, little and thumb in the fixed and free condition. The normal force of middle, ring, little and thumb fingers in the free condition significantly increased ($p < 0.001$) compared to fixed condition. (B) Average tangential force of index, middle, ring, little and thumb in the fixed and free condition. Thumb tangential force in the free condition significantly decreased ($p < 0.001$) compared to fixed condition. Ring and little finger tangential force in the free condition significantly increased ($p < 0.001$) compared to fixed condition. The columns and error bars indicate means and standard error of means.

of the ring and the little fingers to maintain the handle in static equilibrium during the free condition. We found that the normal force of the little finger was not statistically different from the normal force of the ring finger in the free condition (Ring finger: Mean = 2.69N, SD = 0.84; Little finger: Mean = 2.86N, SD = 0.71, t(14) = −0.543, $p = 0.596$, $d_z = 0.14$). By employing the TOST procedure (*Lakens, 2017*), with equivalence bounds of $\Delta_L = -0.93$ and $\Delta_U = 0.93$ for a desired statistical power of 95%, dependent samples of normal forces of the ulnar fingers were found to be statistically equivalent (t(14) = 3.059, $p = 0.00425$). As the observed effect size($d_z = 0.14$) falls within the equivalence bounds, this comparison was deemed to be equivalent.

This was true in both the absolute and % values of the normal forces. The averages (across time, trials, and participants) of normal force and tangential force can be seen in Figs. 2A and 2B.

A two-way repeated-measures ANOVA on average normal force with factors condition and finger showed a significant main effect of condition ($F_{(0.76, 10.64)} = 85.44$; $p < 0.001$, $\eta^2_p = 0.85$) corresponding to a significantly higher ($p < 0.001$) normal force in free condition compared to fixed condition. There was a significant main effect of the finger ($F_{(2.24, 31.36)} = 259.23$; $p < 0.001$, $\eta^2_p = 0.94$) corresponding to a significantly higher

($p < 0.001$) normal force for thumb than other fingers. To check for differences between fingers other than the thumb, we performed a one-way ANOVA. However, we did not find any such difference in the normal force of individual fingers other than the thumb. The interaction condition x finger was significant ($F_{(3.04, 42.56)} = 56.70$; $p < 0.001$, $\eta^2_p = 0.80$) reflecting the fact that the average normal force of thumb in free and fixed condition(9.16N & 5.36N) was significantly higher than the other fingers index(1.65N &1.63N), middle(1.84N &1.33N), ring(2.69N&1.21N) and little(2.86N & 1.09N). The normal force of the little finger (2.86N) in the free condition is significantly greater than the normal force of the index (1.63N, $p < 0.01$), middle (1.33N, $p < 0.001$), ring (1.21N, $p < 0.001$) fingers in fixed condition and index finger (1.65N, $p < 0.05$ ) in the free condition. Ring finger normal (2.69N) in the free condition is significantly greater than the normal force of the index (1.63N, $p < 0.05$), middle (1.33N, $p < 0.01$), ring (1.21N, $p < 0.001$) and little (1.09N, $p < 0.001$) fingers in the fixed condition.

The effects of condition on average tangential force were significant ($F_{(0.91, 12.74)} = 13.44$; $p < 0.01$, $\eta^2_p = 0.5$) according to two-way repeated-measures ANOVA. A significant main effect was found for finger ($F_{(3.8, 53.2)} = 46.87$; $p < 0.001$, $\eta^2_p = 0.77$). This indicated that the thumb tangential force was different from other fingers. Pairwise comparisons showed that the tangential force of the ring and little finger increased significantly ($p < 0.001$) in free condition (1.25N, 1.14N) compared to fixed condition (0.74N, 0.41N). Interaction effects were significant ($F_{(3.64, 50.96)} = 127.54$; $p < 0.001$, $\eta^2_p = 0.90$) for condition x finger reflecting the fact that the average thumb tangential force decreased significantly ($p < 0.001$) in free condition(1.09N) compared to fixed condition(2.70N). The average thumb tangential force in fixed condition (2.70N) is significantly greater ($p < 0.001$) than the average tangential force of index (0.85N, 0.77N), middle (1N, 1.08N), ring (0.74N, 1.25N), and little finger (0.41N, 1.14N) in fixed and free conditions. The average tangential force of little finger in fixed condition (0.41N) significantly decreased than the ring (1.25N, $p < 0.001$) and thumb (1.09N, $p < 0.001$) in free condition, index (0.85N, $p < 0.01$; 0.77N, $p < 0.05$) and middle finger (1N, $p < 0.001$; 1.08N, $p < 0.001$) in both conditions. Ring finger tangential force in free condition (1.25N) is significantly greater than the index finger index (0.85N, $p < 0.01$; 0.77N, $p < 0.001$) in both conditions. In free condition, the tangential force of the little finger (1.14N) is significantly greater ($p < 0.01$) than the tangential force of ring finger in fixed condition (0.74N). Thumb tangential force in free condition (1.09N) is significantly greater ($p < 0.05$) than the ring finger in fixed condition (0.74N).

### Force sharing of the normal forces

Normal force sharing was different between the two conditions. We observed a significant main effect of condition ($F_{(1, 14)} = 13.83$; $p < 0.01$, $\eta^2_p = 1.06$) on the normal force sharing of the individual fingers other than the thumb. Index ($p < 0.001$) and middle ($p < 0.05$) finger contributed significantly lesser normal force share in free condition compared to the fixed condition (see Fig. 3). Further, the percentage of normal force shared by the ring finger was not statistically different from the percentage of normal force shared by the little finger during the free condition (Ring finger: Mean = 30.97%, SD = 8.25; Little
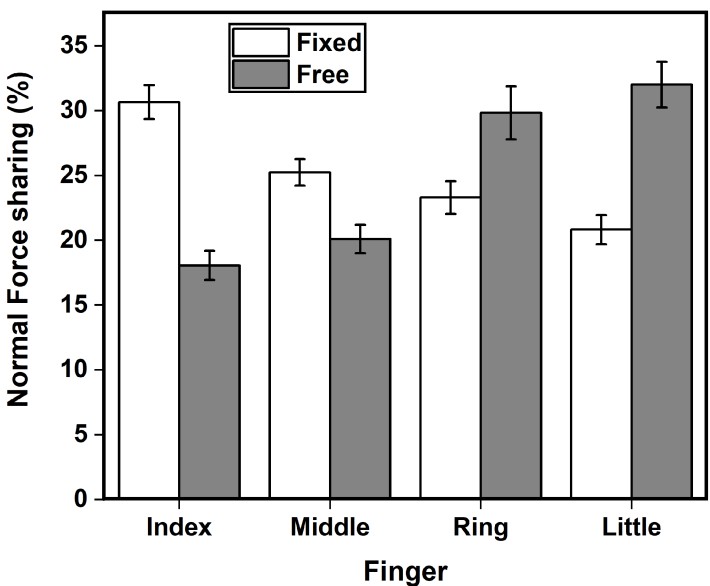

**Figure 3 Normal force sharing in %.** Share of the normal force individual fingers other than the thumb for fixed (white) and the free (grey) condition is expressed as a percentage of total normal force. In the free condition, normal force share of the ring ($p < 0.05$) and little ($p < 0.001$) finger significantly increased compared to fixed condition. Index ($p < 0.001$) and middle ($p < 0.05$) finger showed a reduction in the normal force share in the free condition in comparison to the fixed condition.

finger: Mean = 33.35%, SD = 7.20, t(14) = -0.655, $p = 0.523$, $d_z = 0.16$). However, the TOST procedure confirmed that the comparison was statistically equivalent (t(14) = 2.947, $p = 0.0053$), as the observed effect size was significantly within the equivalence bounds of $\Delta_L = -0.93$ and $\Delta_U = 0.93$.

## Changes in Safety Margin due to condition and fingers

Safety margin changed between the two conditions. A two-way repeated-measures ANOVA was performed using the factors condition and finger. Both factors, condition ($F_{(0.89, 12.46)} = 50.40$; $p < 0.001$, $\eta^2_p = 0.78$) and finger ($F_{(3.44, 48.16)} = 29.26$; $p < 0.001$, $\eta^2_p = 0.67$) showed statistical significance. Post-hoc pairwise comparisons showed significantly higher $SM_z$ for ring finger ($p < 0.05$) and thumb ($p < 0.001$) in the free condition when compared to fixed condition. Interaction effect also showed statistically significant difference ($F_{(3.56, 49.84)} = 66.11$; $p < 0.001$, $\eta^2_p = 0.82$) for the safety margin between the factors. The safety margin of the thumb in the free condition (1.34) is significantly greater ($p < 0.001$) than the safety margin of the index (0.45, 0.55), middle (0.20, 0.35), ring (0.37, 0.56) and little (0.70, 0.66) in fixed and free conditions. Middle finger safety margin during free condition (0.35) is significantly ($p < 0.01$) lower than the little (0.70, 0.66) in both conditions. In fixed condition, middle finger safety margin (0.20) decreased significantly compared to the little finger (0.70, $p < 0.001$ ; 0.66, $p < 0.001$) in both conditions, index (0.45, $p < 0.01$) and ring (0.37, $p < 0.001$) in the free condition, thumb (0.51, $p < 0.01$) in the fixed condition.
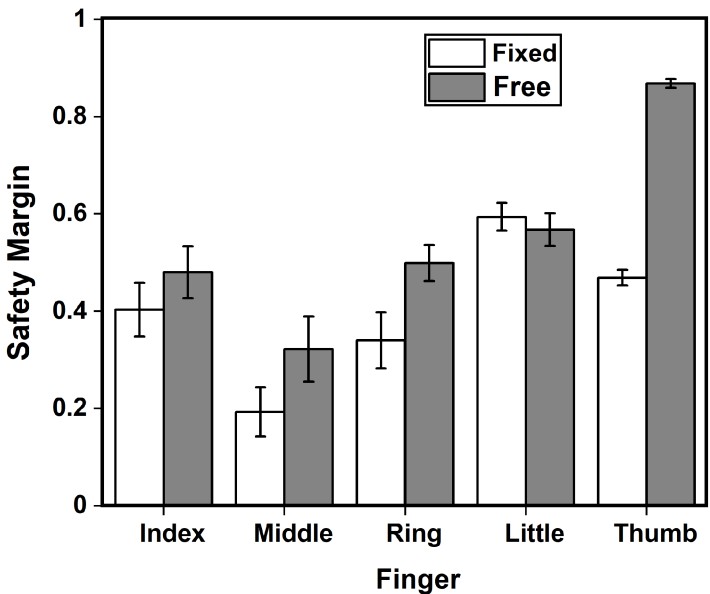

**Figure 4 Safety margin of individual fingers and thumb in fixed and free conditions.** Safety margin of thumb ($p < 0.001$) and ring ($p < 0.05$) finger in the free condition significantly increased compared to the fixed condition. Safety margin of the index, middle and little fingers in free condition were found to be equivalent to the safety margin of the corresponding fingers in the free condition.

In addition to this, the safety margin of index finger (Mean $= 0.45$, SD $= 0.26$) in fixed condition was not statistically different ($t(28) =$ -1.031, $p = 0.311$, $d = 0.37$) from safety margin of index finger (Mean $= 0.55$, SD $= 0.25$) in free condition. Similarly, the safety margin of middle (Mean $= 0.20$, SD $= 0.19$) and little fingers (Mean $= 0.70$, SD $= 0.19$) in fixed condition was not significantly different (Middle: $t(28) =$ -1.796, $p = 0.0833$, $d = 0.64$; Little finger: $t(28) = 0.582$, $p = 0.565$, $d = 0.20$ ) from safety margin of middle (Mean $= 0.35$, SD $= 0.28$) and little finger (Mean $= 0.66$, SD $= 0.19$) in free condition. We observed a statistical equivalence among the safety margin of index ($t(28) = 2.557$, $p = 0.00814$), middle ($t(28) = 1.792$, $p = 0.042$) and little fingers ($t(28) = -3.005$, $p = 0.00277$) between the fixed and free condition as their observed effect sizes falls within the equivalence bounds of $\Delta_L = -1.31$ and $\Delta_U = 1.31$. We confirmed them through TOST $T$-test. These findings are illustrated in Fig. 4.

## DISCUSSION

In our study, participants attempted to maintain the handle in static equilibrium both during the fixed and free conditions. In the fixed condition, when the mechanical constraint was used to restrict the translation of the thumb vertically, the entire load of the handle was shared by the thumb and other fingers. In free condition, the thumb platform was made free to slide over the railing on the handle. The changes in friction that occurred on the surface between the thumb platform and handle can be perceived by the proprioceptors located in the thumb muscles and joints. This sensory information is communicated to

the CNS via the afferent path. In response to that, CNS generates a motor command to the thenar muscles controlling the thumb. The critical difference between conditions is that in the "free condition" the thumb cannot apply desired vertical tangential force as found in the fixed condition. The tangential force of the thumb dropped from ~2.7N to 1N during the free condition. The thumb could only produce 1N force without causing a translation of the slider. If the participant attempts to increase the tangential force above 1N, the thumb platform will slide upwards (violation of experimental instruction).

The drop in the thumb's tangential force caused an increase in the tangential force of virtual finger to overcome the weight of the handle. The tangential force of the virtual finger was ~4.24N which was three times greater than the tangential force of the thumb (1.09N). This, in turn, could cause a tilt of the handle in the counter-clockwise direction. Such a tilt will disturb the rotational equilibrium of the handle. Eventually, there was a compensatory adjustment in the normal and tangential forces of the digits to retain the equilibrium of the handle. According to the mechanical advantage hypothesis, peripheral fingers (index and little) that have larger moment arms (for normal force) tend to produce greater normal force compared to the central fingers (middle and ring) having shorter moment arms during moment production tasks. Earlier, this hypothesis was tested in the pronation or supination moment production tasks to establish static stabilization of the handle when external torques were introduced to the handle. From their results (*Slota, Latash & Zatsiorsky, 2012*; *Zhang et al., 2009*), it was found that the peripheral fingers exert greater normal force. In our study, the applicability of the mechanical advantage hypothesis was examined to check whether the little finger (one among the peripheral fingers) produces a larger normal force compared to the ring finger (having shorter moment arm) to overcome the drop in thumb tangential force. As mentioned earlier, the drop in the thumb's tangential force could cause a tilt of the handle in the counter-clockwise direction. In order to overcome such tilt, clockwise moment (or supination moment) has to be produced by the normal forces of the ring and little finger. Hence, we expected that the little finger would produce greater normal force than the ring finger, causing supination moment in-order to bring the handle back to its equilibrium state.

However, we observed that both the ulnar fingers exerted a statistically equal absolute normal force (in Newton) and normal force sharing (in terms of %). The results of our current study are not in agreement with the mechanical advantage hypothesis as the ulnar fingers exerted comparable normal forces. This might be due to the lesser mass of the handle when compared to the studies (*Shim, Latash & Zatsiorsky, 2005a*; *Gao, Latash & Zatsiorsky, 2006*) which found support for the mechanical advantage hypothesis. Further, in other studies (*Shim, Latash & Zatsiorsky, 2005a*; *Zatsiorsky, Gregory & Latash, 2002*), external torques imposed to the handle was in the range of Newton-meter (Nm). In our study, counter-clockwise tilt caused due to the drop in the thumb's tangential force was in the range of Newton-centimeter (Ncm) which is comparatively lesser. Also, not all the results of a study on finger coordination during the moment production on a mechanically fixed object (*Shim, Latash & Zatsiorsky, 2004*) supported the mechanical advantage hypothesis. One similarity between our study and that of *Shim, Latash & Zatsiorsky (2004)* was that the task involved moment production by the fingers (other than the thumb). However, in our

study, change in thumb forces are not due to the external torque introduced to the handle but rather due to the artificial ("forced") reduction in the thumb contribution to hold the handle. Perhaps, further research is needed with different conditions to check whether there is a monotonic relationship between the finger moment arm and force during various conditions in such tasks involving an artificial reduction in thumb contribution.

The central nervous system probably has chosen to increase tangential forces of ulnar fingers, naturally accompanied by an increase in their normal forces. The increase in normal forces of ulnar fingers would disturb the horizontal equilibrium of the handle (see Fig. 5). Consequently, thumb's normal force increased to 9N (almost doubled compared to thumb's normal force in fixed condition). This helped to balance the forces in the horizontal direction and also to avoid slipping of the thumb slider downwards (*Burstedt, Edin & Johansson, 1997*). We believe that the normal force of the thumb increased in a "feed-forward" manner. A 'non-slip strategy' (*Edin, Westling & Johansson, 1992*) for the thumb by raising the safety margin of the thumb (see Fig. 4) when there was a vertical unsteadiness at the slider platform was chosen by the system. In contrast, if the normal forces of the thumb remain the same or decrease, there will be an increase in the clockwise moment caused due to the ulnar fingers. This would result in the rotation of the handle in the clockwise direction (see Fig. S3). We further examined whether there is any shift in center of pressure (COP) of the fingers and thumb in the vertical direction to compensate the tangential force drop in the thumb. So, we performed a planned pairwise comparison on the difference in COP shift from the initial to the final point between the conditions. We found that there was not any significant difference in all the fingers and thumb.

By definition, the safety margin is meant to be the amount of extra normal force applied to avoid slipping. From the results of our study, we could observe a significant increase in the safety margin of the ring finger as there was a rise in the normal force of ring finger to compensate for the drop in the tangential force of the thumb. It is known that the tangential force of the virtual finger increases to compensate for the drop in the tangential force of the thumb in order to preserve the vertical equilibrium of the handle. Prior studies (*Radwin et al., 1992*; *Kinoshita, Murase & Bandou, 1996*; *Zatsiorsky, Li & Latash, 1998*; *Aoki, Francis & Kinoshita, 2003*) have shown that the index and middle fingers play a major role in producing greater forces compared to the ring and little finger. Next to the thumb, the index finger is considered to contribute in a great way for the independent force control (*Jones & Kamper, 2018*). Meanwhile, the middle finger (one among the central finger) was responsible for supporting the weight of the handle (*Zatsiorsky, Gregory & Latash, 2002*). Therefore, we speculated that the index and middle finger would share a greater tangential force compared to the ring and little finger during the multi-finger prehension task (see Fig. 6). However, it cannot be accompanied by the greater share of normal forces as it could cause a further tilt in the counter-clockwise direction. Hence, we expected that there would be a drop in the normal force of the index and middle finger in the free condition. We considered this as the first option (see Fig. 6B). Alternatively, it is possible that the normal and tangential forces of the radial fingers would decrease while the forces of the ulnar fingers would increase to compensate for the drop in the tangential force of the thumb (see Fig. 6C). Finally, as the third option, it is possible that there will be no changes in the
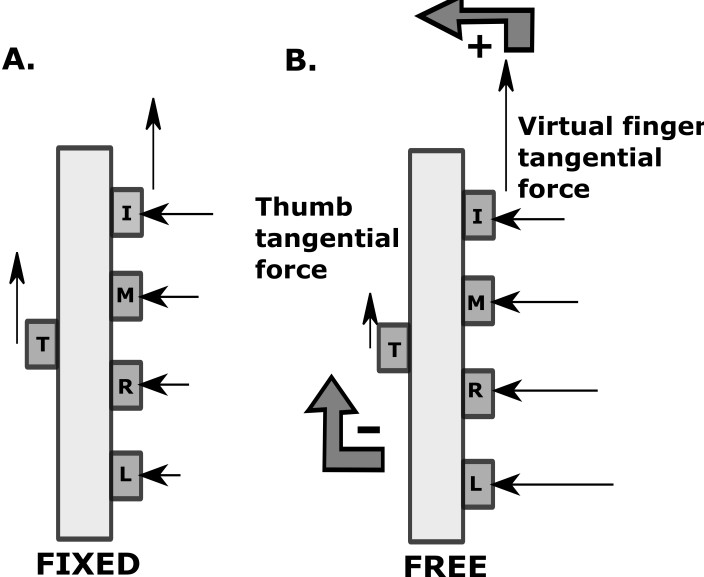

**Figure 5  Force and moment distribution pattern in fixed and free condition.** The length of the arrow approximately corresponds to the absolute magnitude of the force. In fixed condition, moment due to normal force of virtual finger is approximately zero and moment due to normal force of thumb is minimal, so both are not represented. Note the magnitude of virtual finger tangential force and thumb tangential force remained almost the same in the fixed condition. In free condition, '+' sign indicates anti-clockwise moment and '-' sign indicates clockwise moment. The decrease in the tangential force of thumb and an increase in the virtual finger tangential force is shown.

normal and tangential forces of radial fingers, but both normal and tangential forces of ulnar fingers increase significantly when compared with the fixed condition (see Fig. 6D). We observed that our results were similar to the third option.

According to our second hypothesis, there will be a significant drop in the safety margin of the index and middle finger during the free condition compared to fixed condition. Contrary to our expectations, the safety margin of the index and middle finger showed no statistical difference between the fixed and free conditions. Hence, our findings were not in agreement with the second hypothesis, as well. One question that arises here is whether we can define safety margin in this case since the slider was frictionless, and by definition, safety margin does not exist when friction is zero. Friction was minimal (non-zero) between the slider and the handle due to the presence of ball bearings. Friction was much higher between the thumb and the ATI Nano sensor (~0.97, consistent with other studies). If any slip happens, it would first happen at the interface of the slider and the railing, not the sensor and finger. The instruction was to hold the thumb platform in position by matching the horizontal line on the thumb platform to the horizontal line drawn at the midline between the middle and ring finger. The participants followed the instruction without causing any vertical sliding of the thumb platform. Therefore, the safety margin of the thumb is defined and increased in the free condition.

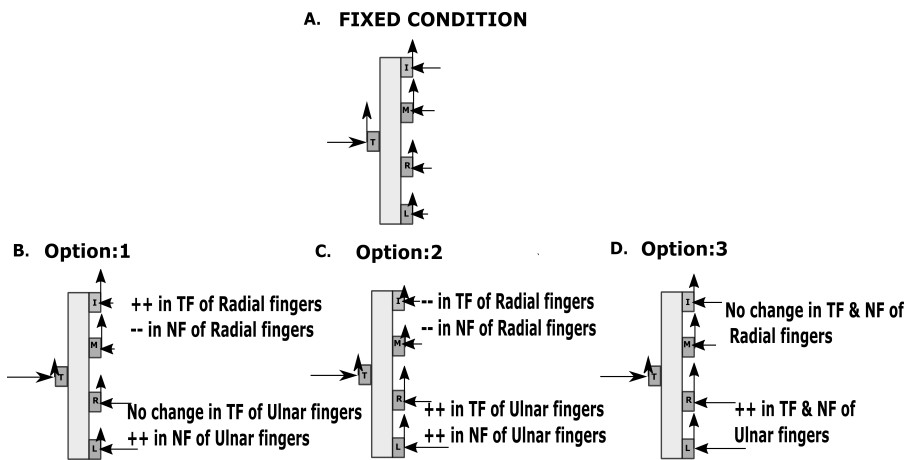

**Figure 6** **Different possibilities of changes during free condition.** Three different possibilities that we could expect when there is a change in the force of the thumb due to the artificial reduction of thumb contribution to holding the object. The length of the arrow signifies the magnitude of the force exerted. (A) Force distribution during the fixed condition (B). Option 1 involves in raising the normal forces of ulnar fingers and tangential forces of radial fingers while decreasing the normal forces of radial fingers. (C) Option 2 involves increasing the normal and tangential forces of ulnar fingers while decreasing the normal and tangential forces of radial fingers. (D) Option 3 involves increasing both normal and tangential forces of the ulnar fingers and producing no change in the forces of radial fingers. NF refers to Normal force and TF refers to Tangential force. Sign '++' refers to increase, and '- -' refers to decrease.

Due to the unsteady thumb platform, proprioceptors on the thumb detect the frictional change under the slider. The CNS, in turn, responds by necessitating the mechanical action of lowering tangential force of the thumb. The magnitude of thumb tangential force in free condition depended on the mass of the thumb platform. We consider this to be the first local change which initiated the synergic effect in the ring and little finger. In our current study, the following chain effect was observed during the free condition: mechanical constraint to fix the thumb in position was removed → tangential force of the thumb decreases → VF tangential force increases → rise in the counter-clockwise moment → compensated by the increase in the normal force of ring and little finger → moment due to normal force of ring and little finger increases in the clockwise direction to counterbalance the rise in the counter-clockwise moment → increase in the normal force of ring, and little fingers were compensated by the increase in the thumb's normal force. Thus, the change in the tangential force of the thumb resulted in the re-arrangement of normal force of all the fingers. This is evident from Fig. S2, where we observe an increase in the percentage change in the normal forces of all the fingers and thumb in free condition compared to the fixed condition. On the other hand, the percentage change in the tangential force of thumb decreased in the free condition. The local change of drop-in thumb tangential force results in the tilting of the handle in the counter-clockwise direction (*Niu, Latash & Zatsiorsky, 2007*). Meanwhile, synergic change of increasing the normal forces of the ulnar fingers brings about a compensatory tilt in the opposite direction. Therefore, it helps the handle to retain its rotational equilibrium. The normal and tangential force adjustment at thumb

was primarily due to a choice made by the controller driven by task mechanics and task instruction.

Comparable normal forces exerted by the two ulnar fingers may be attributed to at least two distinct factors: biomechanical constraints and neural interdependency. Biomechanical constraints include the mechanical linkages and the tendinous interconnections. Firstly, the main source of the gripping force in the ulnar fingers was produced by the Flexor Digitorum Profundus (FDP). The tendons of the FDP muscle extend from the forearm to the tip of the index, middle, ring and little fingers. These tendons are responsible for flexion of the distal interphalangeal joint to exert appropriate normal forces. Since the tendons of middle, ring and little fingers share a common muscle belly, at the forearm level, contraction of one portion (or compartment) of the FDP muscle could cause shortening of the neighbouring muscle compartment of the same FDP. Thus, the mechanical linkage between the muscle compartments of FDP, restrict the independent rise in the normal force of the target finger. Secondly, at the palmar level, it is found that there is a tough fibrous sheet that interconnects the tendons of the middle, ring and little fingers (*Riehle & Vaadia, 2005*). Therefore, during the free condition, the flexion of the little finger (to cause an increase in the normal force of little finger) to overcome tangential force drop in the thumb would be accompanied by the flexion of the adjacent ring finger probably as a result of interconnection between the tendons.

Apart from the biomechanical effects, there also exists an overlap in the motor units territories of the ring and little fingers at the medial portion of the FDP muscle which is responsible for the flexion of ulnar fingers (*Kilbreath & Gandevia, 1994*). In a study conducted by *Kilbreath et al. (2002)*, during the weak voluntary grasping of a cylindrical object, spike-triggered average (STA) forces were measured under each finger. It was shown that the STA forces measured under the ring finger due to the activity of little finger motor units was almost two-thirds of that force produced under the little finger due to the little finger motor units. This change in force caused by the little finger motor units under ring finger was significantly greater ($p < 0.001$) than the change in force produced by index, middle, ring motor units under their adjacent fingers.

As there is a necessity to compensate for the drop in the clockwise moment caused by the tangential force of thumb, little finger motor units get activated (because of the longer moment arm of the little finger) to cause flexion (or increase in normal force) of the little finger. Subsequently, perhaps this results in the activation of the little finger motor units found at the ring finger portion of FDP muscle. Apart from the increase in the normal force of the ulnar fingers due to the little finger motor units, the normal force of ring finger increases due to the activation of ring finger motor units. Hence, this distinct behaviour of the activation of little finger motor units (on small and ring finger portion of FDP muscle) and ring finger motor units on ring finger portion of FDP could be a reason for the exertion of significantly comparable normal force by the ulnar fingers.

## CONCLUDING COMMENTS

When the thumb undergoes a local change of decrease in tangential force, handle equilibrium is disturbed. Subsequently, the handle's equilibrium was restored by increasing

the normal force of the ring and little finger. It was evident from our current study that the ring and little finger normal force showed a statistically comparable increase to counteract the drop in the thumb tangential force. Thus, the little finger (one of the peripheral fingers) did not produce a greater share of normal force than the ring finger to cause supination moment for balancing drop in thumb tangential force. Therefore, our result does not support the mechanical advantage hypothesis, at least for the specific mechanical condition considered by us. In addition to this, there was no statistical difference in the safety margin of the index and middle finger in free condition compared to the fixed condition. This study can also be extended with various conditions that involve in examining mechanical advantage hypothesis in different situations. Further research is required to clarify how well these results generalise to other conditions. Our future studies will focus on investigating the hypothesis with various tasks that encompass different magnitudes of mechanical requirements.

## ACKNOWLEDGEMENTS

We thank Niraj Kumar and Archana MS for their contribution to the data collection part of the experiment. The authors thank the members of Neuromechanics lab, IIT Madras for their critical inputs on an earlier version of the manuscript

### Funding

The Department of Science & Technology, Government of India, supported this work, vide Reference Nos SR/CSRI/97/2014 & DST/CSRI/2017/87 under Cognitive Science Research Initiative (CSRI) (awarded to Varadhan SKM). The funders had no role in study design, data collection and analysis, decision to publish, or preparation of the manuscript.

### Grant Disclosures

The following grant information was disclosed by the authors:
Department of Science & Technology, Government of India.
Cognitive Science Research Initiative (CSRI): SR/CSRI/97/2014, DST/CSRI/2017/87.

### Competing Interests

The authors declare there are no competing interests.

### Author Contributions

- Banuvathy Rajakumar performed the experiments, analyzed the data, prepared figures and/or tables, authored or reviewed drafts of the paper, and approved the final draft.
- Varadhan SKM conceived and designed the experiments, performed the experiments, authored or reviewed drafts of the paper, acquired funding for the study, and approved the final draft.

## Human Ethics

The following information was supplied relating to ethical approvals (i.e., approving body and any reference numbers):

The Institutional Ethics Committee of the Indian Institute of Technology Madras approved this research (IEC/2016/02/VSK-2/12).

## Data Availability

Data are available on Figshare: Rajakumar, Banuvathy; SKM, Varadhan (2019): Comparable behaviour of ring and little fingers due to an artificial reduction in thumb contribution to hold objects. figshare. Dataset. https://doi.org/10.6084/m9.figshare.10738067.v2.

## Supplemental Information

Supplemental information for this article can be found online at http://dx.doi.org/10.7717/peerj.9962#supplemental-information.

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
