# Peer review of "Comparable behaviour of ring and little fingers due to an artificial reduction in thumb contribution to hold objects"

_PeerJ, doi:10.7717/peerj.9962_

## Round 0.1 · original submission · Major Revisions

The two reviewers and I see much merit in this paper, but have some comments we think will improve the clarity of the paper.

·

Basic reporting

The manuscript is well written and the ideas are clearly expressed and developed. The experimental design introduces a mechanism that is able to minimize the contribution of the thumb tangential force in the holding task and induces a chain effect of force adjustment in other fingers. However, the conclusion was mainly driven by the observation of statistically equal normal forces exerted by ring and little fingers when the tangential force of the thumb cannot be translated from the force sensor to the object due to the frictionless interface. It will strengthen the argument if the authors discuss some factors that could contribute to equal finger force production.

Experimental design

The experiment measures individual finger force distribution in a five-digit grip to hold task under two different conditions. A minor concern is that the safety margin of the thumb in the Free condition violates the definition and cannot be computed since there is no margin of no-slip for a frictionless surface.

Validity of the findings

Compared to the previous studies, this present paper compares the finger force distribution only among two essentially different objects, i.e. different tasks. Without a comparable mechanical requirement of individual finger force, it is difficult to determine whether a finger is more mechanically advantageous than another finger. In order to falsify the MAH, the authors might want to compare tasks that only scale up or down the mechanical requirements.

Additional comments

In general, this manuscript presents clearly the methods and results with a discussion of a prehension behavior limited to the MAH and safety margin. My suggestion is to include discussion of some other factors, such as finger biomechanical constraints or neural interdependency.

·

Basic reporting

In general, the experiments are presented clearly and the results are easy to follow. Sufficient relevant background was provided, and cited appropriately. Some of the results require clearer presentation (in particular figure 2), and some of the figures could be removed. The raw data were provided using figshare and appear to be complete, although an explanation of the columns would be useful. The paper is appropriately structured.

Experimental design

The research is original, although the research question is not well motivated - novelty is not a strong motivation for performing a study. The technical details are presented and it is clear that the work was performed to a high technical standard. Appropriate ethical approval was received. The methods were described with sufficient detail to understand how the experiment was performed. The description of the "two one sided t-tests" needs to be expanded.

Validity of the findings

The results of the study are clearly and thoroughly presented, and all relevant data has been presented. The discussion is well tied to the data, apart from the claims that the study "falsifies" the mechanical advantage hypothesis. In this study, the authors compared the normal and tangential forces applied in two conditions when grasping a handle with 5 fingers, with the thumb in the middle vertically 1) with no additional constraints 2) when the tangential (vertical) force is limited. They describe the changes in the strategy of choosing the distribution of forces caused by the change in requirements. In particular, they observe that given the increased moment required to be generated by the "virtual finger" (fingers 2 to 5), the little finger and ring finger play similar roles. They note that their results are in contrast to the mechanical advantage hypothesis, which suggests that the little finger should apply a greater force (due to its longer moment arm). However, the conclusions regarding "falsifying" the mechanical advantage hypothesis do not seem to be sufficiently supported by the data, nor is any alternative hypothesis presented.

Additional comments

Page 6, line 38: remove "in"
Line 39: "the index finger"

It would be useful in the abstract to say why the mechanical advantage hypothesis exists (i.e. what is the benefit)

Page 7, Line 72: "the hand"

Line 74: "it is known . .. " - it would be helpful to explain why the forces change due to these perturbations
Line 82: you should explicitly state that they have longer moment arms if the fingers are spread out evenly on the handle

Page 8, line 114: "for the moment caused due to the vertical unsteadiness of the thumb slider". Isn't it rather a change in moment caused by the reduced normal force when performing a secondary task?
Line 116: "By this way" -> "In this way"

Line 117: "we artificially reduced the contribution of thumb in holding the handle". As this is a critical part of your paper, please elaborate. Reduced normal force? Tangential force? Why / how does this change the moment produced by the thumb?

Line 123: At this point, a diagram of the different possibilities we could expect as a result of the change in thumb force would be helpful

Line 125: virtual finger should be defined

Page 9, line 155: If the friction of the mechanism is very low, it is not clear to me how they could hold the handle with the thumb and apply a vertical force without the handle sliding

Page 13, line 278: If you use the two one-sided t-test approach, you should specify what are the bounds you are using

Line 294: provide SD for the angle values. Please provide t / p values for the statistics presented, including the equivalence tests.

Line 304: "normal force produced by the index finger was found to be non-different". Do you mean that the variance of the normal force was low throughout the trial? How did you quantify this? I don't understand how this relates to "overlapping"

Line 324: You present "The interaction condition x finger was significant", but then only support this by differences found in both conditions. The interaction needs to be explained by different differences in the two conditions

Line 365: LDA analysis - I'm not sure what the purpose of the LDA analysis is. It seems obvious that different strategies are used in the two conditions, and I am not sure what the LDA analysis adds to this.

Page 15, line 387: "The thumb only has to support the slider platform whose weight is around 1N". The main difference between conditions is not that it "only" has to support the slider but rather than it cannot apply the "desired" vertical force (as was used in the fixed task)

Page 16, line 413: As you noted in the introduction, "the moment arm of the finger had a monotonic non-linear relationship with the force produced by that finger". This seems to be able to provide more evidence for the MAH, whereas in your study there were only two levels considered, and so it is harder to determine whether your findings negate the MAH. You should also explicitly note that the change in forces here are not due to an external moment but rather a (forced) change in finger forces

Page 18, line 502 "falsified the mechanical advantage hypothesis", this seems overly strong. "Does not support" would be more appropriate

Figure 2: In (a), where is the fixed index force? In some of the panels in (b), it is not possible to distinguish between the SDs. Also, why does the dashed line stop / start midway?

Figure 2 + 3 do not seem to present the same data. It seems like Figure 3 should be the average of Figure 2, but this does not seem to be the case

Figure 2-4 are essentially presenting the same data, less figures would improve the clarity of the paper

---

## Round 0.2 · Minor Revisions

On behalf of the two reviewers, I thank you for your hard work in addressing the initial concerns of this manuscript. The two reviewers and I are happy to accept this paper for publication, pending some very minor comments from reviewer two.

·

Basic reporting

no comment

Experimental design

no comment

Validity of the findings

no comment

Additional comments

This revision addressed all my concerns.

·

Basic reporting

No comment

Experimental design

No comment

Validity of the findings

No comment

Additional comments

The authors have corrected most of my concerns with the paper. Some minor comments:

- Abstract: "falsify" - similar to the change you made in the manuscript, I would also remove / change the word "falsify" in the abstract to a something like "does not support"

- Figure 6 is useful for understanding the possible changes - in the text, please note which option (A,B or C) matches your prediction, and which matches the outcome observed in the experiment

---

## Round 0.3 · accepted · Accept

I thank the authors for the dedication and attending to the reviewers comments. I would now recommend this manuscript be accepted for publication in PeerJ.